# Effect of HHP and UHPH High-Pressure Techniques on the Extraction and Stability of Grape and Other Fruit Anthocyanins

**DOI:** 10.3390/antiox12091746

**Published:** 2023-09-10

**Authors:** Antonio Morata, Juan Manuel del Fresno, Mohsen Gavahian, Buenaventura Guamis, Felipe Palomero, Carmen López

**Affiliations:** 1enotecUPM, Department of Chemistry and Food Technology, ETSIAAB, Universidad Politécnica de Madrid, 28040 Madrid, Spain; juanmanuel.delfresno@upm.es (J.M.d.F.); felipe.palomero@upm.es (F.P.); carmen.lopez@upm.es (C.L.); 2Department of Food Science, National Pingtung University of Science and Technology, Pingtung 91201, Taiwan; mg@mail.npust.edu.tw; 3Centre d’Innovació, Recerca I Transferència en Tecnologia Dels Aliments (CIRTTA), TECNIO, XaRTA, Departament de Ciència Animal I Dels Aliments, Facultat de Veterinària, Universitat Autònoma de Barcelona, 08193 Bellaterra, Spain; buenaventura.guamis@uab.cat

**Keywords:** anthocyanins, HHP, UHPH, grape, wine, clean labels

## Abstract

The use of high-pressure technologies is a hot topic in food science because of the potential for a gentle process in which spoilage and pathogenic microorganisms can be eliminated; these technologies also have effects on the extraction, preservation, and modification of some constituents. Whole grapes or bunches can be processed by High Hydrostatic Pressure (HHP), which causes poration of the skin cell walls and rapid diffusion of the anthocyanins into the pulp and seeds in a short treatment time (2–10 min), improving maceration. Grape juice with colloidal skin particles of less than 500 µm processed by Ultra-High Pressure Homogenization (UHPH) is nano-fragmented with high anthocyanin release. Anthocyanins can be rapidly extracted from skins using HHP and cell fragments using UHPH, releasing them and facilitating their diffusion into the liquid quickly. HHP and UHPH techniques are gentle and protective of sensitive molecules such as phenols, terpenes, and vitamins. Both techniques are non-thermal technologies with mild temperatures and residence times. Moreover, UHPH produces an intense inactivation of oxidative enzymes (PPOs), thus preserving the antioxidant activity of grape juices. Both technologies can be applied to juices or concentrates; in addition, HHP can be applied to grapes or bunches. This review provides detailed information on the main features of these novel techniques, their current status in anthocyanin extraction, and their effects on stability and process sustainability.

## 1. Introduction

Anthocyanins are flavonoid pigments with variable colors from yellow-orange (490 nm) to red-blue (540 nm), with healthy nutritional properties widely distributed in fruits and flowers and with useful properties as food pigments [1,2,3,4,5]. Anthocyanins can act as antioxidants, phytoalexins, or antimicrobial compounds [6,7]. Anthocyanins behave as weak acids, hard and soft electrophiles, nucleophiles, and metal ion binders [8]. The main degradation effects that can affect anthocyanins are oxidative yield processes, which can be accelerated by temperature [9]. Anthocyanins can also be affected by light (photooxidation) [10] and oxidized by oxygen directly or via enzymes such as polyphenol oxidases and peroxidases [9,10]. Stability and color are also affected by pH, SO_2_ bleaching, copigmentation, or the formation of polymers with other flavonoids, such as catechins or proanthocyanidins (Figure 1). Non-thermal emerging technologies such as high-pressure processing by High Hydrostatic Pressure (HHP) or Ultra-High Pressure Homogenization (UHPH) can be used to gently extract and preserve juice anthocyanins, increasing extraction rates and stability [11,12].

High Hydrostatic Pressure (HHP) is the application of pressure to food by a pressurizing fluid, usually water, inside a highly resistant steel vessel (Figure 2). Initially, the vessel is filled with water using a high-flow, low-pressure pump, and later, the water is pressurized to a typical range of 500–600 MPa using a special pump called a pressure intensifier (Figure 2). Under these pressure conditions, the water and the food inside are compressed by about 4%. The pressurization is produced with some temperature increase due to adiabatic heat compression, which is typically lower than 4 °C/100 MPa (<3.4 °C water, <3.8 °C orange juice, [13]), producing a global heating of up to 24 °C at 600 MPa. The temperature increase also depends on the composition, which is higher for oil components. HHP is a technique that can be considered a non-thermal technology because of this soft heating, without any degradative effect on thermally sensitive molecules such as pigments or aromatic compounds [14,15]. Low adiabatic heating can be additionally controlled by cooling the vessel with heat exchangers.

HHP, unlike thermal treatments, does not have enough energy to affect covalent bonds; therefore, small molecules with sensory impact are protected, and HHP can be considered a gentle technology. The formation of unexpected molecules has not been observed during HHP processing [16]. Furthermore, HHP is a pressure process produced in a closed vessel where aeration and oxidative processes are minimized, thus reducing damage to anthocyanins. 

Many studies have shown that using HHP at room temperature preserves nutritional value and has a very low effect on the levels of anthocyanins in fruits and vegetables [17]. The HHP-treated juices show good color stability, better phenolic content, and higher antioxidant activity [18]. 

Ultra-High Pressure Homogenization (UHPH) is a high-pressure technology in which a fluid is pumped at more than 200 MPa (typically 300 MPa) through a thin, high-resistance steel pipeline and later depressurized via a special valve made of a high-resistance alloy [19,20]. The valve is often made of tungsten carbide and is usually coated with nanolayers of extremely resistant carbon polymers. Inside the valve, fluids are subjected to extreme shear forces and impacts, causing nanofragmentation of microorganisms, colloids, and biopolymers to 100–500 nm [20,21] (Figure 3). 

Some interesting effects of submicron nanofragmentation are the elimination of microorganisms, inactivation of enzymes (including oxidative polyphenol oxidases, PPOs), increase in antioxidant activity, and better colloidal stability [20,21,22,23]. Additionally, UHPH is very gentle with sensory quality. No degradation of terpenes [21], anthocyanins [22], or thermal markers such as hydroxymethyl furfural was observed during the processing of grape juices [21].

## 2. Main Features of HHP and UHPH 

High-pressure techniques can be considered gentle, non-thermal technologies with specific features to preserve the nutritional and sensory quality of food products. However, they have several specifications regarding the processing conditions (pressure, residence time, and temperature), the requirements of the products to be processed (particle size and liquid/solid), the antimicrobial capacity (pasteurization or sterilization), the inactivation of enzymes, and the effects on molecules with sensory impact (Table 1). Microbial inactivation in HHP depends on the pressure and residence time, with maximum efficacy being achieved at 500–600 MPa for 1–10 min [16]. In UHPH, it is also dependent on pressure, valve design, and in-valve temperature, with better results obtained at 300 MPa and sterilization at in-valve temperatures of 140–150 °C [19,20].

UHPH has the advantage of being continuous [20] with high antimicrobial [19] and antioxidant effects [21,23,24] due to the efficient control of oxidative enzymes (PPOs). However, it can only be used for liquids (grape juice) with a particle size of less than 500 µm. Although high temperatures can be generated in the valve by mechanical effects, often higher than 75 °C, UHPH can be considered a non-thermal technology because, at 300 MPa, the fluid flows in the valve at Mach 3 for less than 0.2 s, which is insufficient for thermal degradation of small molecules with sensory impact. After the valve, the intense nanofragmentation reduces the average size of the colloidal particles as a function of pressure and temperature. The dimensions have been measured in grape juice by laser diffraction and atomic force microscopy, giving values in the range of 100–400 nm [21] and 235–744 nm (average 457 ± 140 nm), respectively [22].

HHP is a discontinuous or batch process but can be applied to whole grapes, bunches, or grape pomace [25,26]. The effect of HHP on enzymes is variable. In the case of PPOs, which can strongly affect the stability of anthocyanins, inactivation depends on the conditions (pressure, time, and temperature) [27]. For example, in raspberry and strawberry food products, at least 400–600 MPa and 5–10 min at room temperature are required [28,29,30].

Industrial scale-up of high-efficiency UHPH systems is now possible with existing UHPH pumps capable of continuous operation at 300 MPa or higher with a pressure imbalance of 1 MPa (<1%). UHPH pumps are available from 60 L/h to 10,000 L/h, covering the range from pilot systems to industrial devices (https://www.ypsicon.com/ (accessed on 24 August 2023)). The special high-strength tungsten carbide valves with ceramic or carbon nano coverings can be scaled up to several of these flow ranges. The electric motor of UHPH pumps, which is the most important consumption range, is 11–18 kW for pumps of 60–150 L/h (personal communication, Ypsicon, Barcelona, Spain).

There are currently several companies (Hiperbaric (Burgos, Spain), Avure (Chicago, IL, USA), Uhde (Hagen, Germany), Kobelco (Livonia, MI, USA) and others [16]) working on industrial HHP systems capable of operating at up to 600 MPa with vessels ranging from 55 to 525 L suitable for processing 270–3210 kg/h of food products (https://www.hiperbaric.com/en/hpp-technology/ (accessed on 24 August 2023)). There are also semi-continuous bulk HHP systems for liquids able to work at over 4000 L/h ([16], https://www.hiperbaric.com/en/hpp-technology/equipment/hpp-in-bulk/ (accessed on 24 August 2023)).

**Table 1 antioxidants-12-01746-t001:** Main features of UHPH and HHP.

Characteristics	HHP	References	UHPH	References
Mode	Batch or semi-continuous	[13,14,16]	Continuous	[19,20,31]
Temperature	−20–60 °C		70–160 °C	
Pressure range Optimal pressure	200–600 MPa 500–600 MPa		200–600 MPa 300 MPa	
Residence time	2–10 min		<0.2 s	
Increment of temperature during processing	<4 °C/100 MPa		70–90 °C	
Size requirements	Smaller than the vessel diameter		Particle size < 500 µm	
Antimicrobial	Pasteurization Sterilizing variable if T > 100 °C	[32]	Sterilizing if T > 140 °C	[20]
Enzymes	Variable Sometimes activation at low pressure (200 MPa) and inactivation at 400–600 MPa In some conditions, suitable inactivation of PPOs ^1^	[16,27,32,33]	Inactivation, highly effective of PPOs ^1^	[21,23,24,34]
Terpenes	unaffected	[14,15,17,25]	unaffected	[21,22,23]
Thiols	unaffected		unaffected	
Anthocyanins	Protected		Protected	
Pyranoanthocyanins	Higher formation of Vitisin A	[35,36]	Similar formation during fermentation of grape juice processed by UHPH	[22]
Polymeric anthocyanins	Formation in some conditions in model solutions	[35]	Unknown	

^1^ PPOs: Polyphenol oxidases.

Both technologies have been proposed to replace or reduce the SO_2_ content in grapes and wines due to their antimicrobial and antioxidant properties [21,23,25]. In the case of grapes or whole bunches, they can be processed directly without packaging and, after depressurization, destemmed, crushed, and pressed to obtain a juice with a higher concentration of anthocyanins and other compounds such as phenols or aroma. Alternatively, the crushed grape can be pumped into a tank for red winemaking to ferment the juice. 

## 3. Extraction of Anthocyanins by HHP

The extraction of anthocyanins by HHP is influenced by the intensity of pressure, temperature, and polarity (ethanol content) [26]. In addition, the substitution pattern in the anthocyanin ring also affects the extraction, depending on the -OCH_3_ and -OH substituents [26]. We have observed that after HHP treatment of grapes, even if the external appearance remains unaffected, perhaps just a little more brilliant, the surface looks as if it has been pored; the migration of anthocyanins into the pulp can be observed because of the red color (Figure 4) [25]. The permeability of the cells increases, and the diffusion of anthocyanins is improved [37]. The anthocyanins are released from the vacuoles in the skins and later migrate to the pulp, but also to the seeds during the HHP treatment, and the effect is pressure-dependent. The higher the pressure, the more intense the extraction, reaching higher concentrations of anthocyanins in the juice of the grapes processed at 550 MPa/10 min compared to the 200 and 400 MPa treatments [25]. A similar effect was observed in total polyphenols but with a less pressure-dependent behavior when a comparable extraction was observed at pressures in the range of 200–550 MPa. The increase in the extraction of total polyphenols was in the range of 20–25% [25]. The extraction of anthocyanins can range from 20 to 80%, depending on the processing parameters (pressure, time, temperature, and polarity) and the conditions of the grape (ripeness, skin thickness and resistance, and previous processes) or grape by-products [25,26,34]. 

HHP has shown a protective effect on several anthocyanins with low degradation after treatment (Table 2) and protection of antioxidant capacity [33,38,39]. The protective effect on anthocyanins is better when pre-processing is carried out at low oxygen levels [40]. Inertization by a nitrogen atmosphere in the pre-processing of berries probably reduces the effect of residual PPO activity on anthocyanin oxidation. Color retention is usually better in HHP treatments than in conventional thermal processing (80–90 °C) [41], probably due to the intensification of oxidation. 

The protective effect on anthocyanins in preserving the antioxidant capacity and the partial inactivation of PPOs make HHP a gentle technique for extracting anthocyanins from fruits or vegetable tissues. The subsequent stability of the extracted anthocyanins under refrigeration favors the stability and valorization of these pigments to be used in the food industry, obtain wines with faster macerations, or even fermentations in the absence of skins, facilitating winemaking.

## 4. Extraction of Anthocyanins by UHPH

Anthocyanins are essential for the color, antioxidant properties, and sensory quality of berry juices and their fermentative products, such as wine. After juice extraction, anthocyanins can remain entrapped or adsorbed in colloidal particles of the pulp and are also more exposed to the action of PPOs than in the whole berry. Mechanical action and nanofragmentation in the UHPH valve are powerful tools to release the anthocyanins from the colloidal fragments and to avoid the effects of PPOs by inactivating them.

In the application of UHPH, the juice is pressurized by pumping with a special pump that works at >200 MPa (usually at 300 MPa); the capillary pipeline section is smaller than 1 mm so that any fluid with a colloid size higher than 500 µm cannot be processed [20]. Grapes or raw grape skins cannot be processed directly because of the previous size limitation, but UHPH can be applied to the unsettled must obtained by pressing after cold soaking or enzyme treatments of the crushed grape [22]. It would also be possible to mechanically process the skins with the pomace using a milling system and later apply UHPH to the puree.

A high efficiency of UHPH to disaggregate and destroy biofilms by mechanical effects has been observed [48]. The processing of plant matrices by UHPH produces a better and more stable colloidal structure [49], which can be measured by particle settling produced by centrifugation [50]. In addition, higher viscosity and turbidity can be observed due to the formation of stable colloidal nanoparticles [51]. What can be observed after the UHPH processing at pressures higher than 200 MPa is the nano-fragmentation of all the microorganisms and colloidal fragments from the plant tissues (Figure 5). In the raw juice of unsettled Tempranillo grape (*Vitis vinifera* L.), yeast cells and large colloidal fragments can be observed (Figure 5A), which are nano-fragmented during UHPH by the impact and shear forces generated in the valve and a thinner and more regular colloidal structure can be observed (Figure 5B–D). In addition, the colloids are slightly colored red by the anthocyanins (Figure 5A (Tempranillo), Figure 5E (Cabernet sauvignon)), which disappear from them in the UHPH-treated juice (Figure 5B–D,F) [22]. Using Atomic Force Microscopy (AFM), the average colloid size in Cabernet sauvignon juice before and after UHPH treatment was measured to be 1.32 ± 0.46 µm (predominantly super-micron scale) and 0.46 + 0.14 µm (nanometric scale), respectively [22]. Previously, and in agreement with this value, a similar size range of 100–400 nm was measured in grape juice treated with UHPH using laser light scattering [21]. Nanofragmentation at a size greater than 100 nm is relevant because lower values may have nanosafety implications due to the possibility of crossing plasmatic barriers.

Applying UHPH produces a highly effective nano-fragmentation of the plant tissues and cells, releasing the anthocyanins from them and facilitating high and rapid extraction (Figure 5) [22]. Therefore, the application of UHPH is a powerful, gentle technology not only to eliminate microorganisms but also to release pigments [22], thus protecting the color [51,52] and other nutrients, antioxidants, and nutraceutical compounds [51,52,53,54,55].

Several significant works have been carried out on the use of High-Pressure Homogenization (HPH) or UHPH for the processing of juices or plant-based beverages [21,23,34,49,50,51,56,57,58], but not too much in the processing of red grape juice or berries rich in anthocyanins. However, there are some recent works at 150–300 MPa with important results, including the release of anthocyanins from plant cells or fragments, the protection of color and the control of browning, the specific effect on acylated anthocyanins, the inactivation of PPOs and the preservation of antioxidant activity (Table 3) [22,59,60], together with effective control of microbial loads. Conversely, a 33–38% reduction in anthocyanins was observed in mulberry juice processed at 200 MPa with a pilot UHPH system at a flow rate of 10 L/h, which the authors associate with uncontrolled peroxidase or polyphenol oxidase, but which can be avoided with ascorbic acid [61]. Most of the studies reporting a reduction in anthocyanin content work at lower pressures (50–200 MPa) and in multi-cycle mode (2–5 passes) [60,62,63], which increases the possibilities of oxidation and damage by mechanical effects during several passes through the UHPH valve. Furthermore, at a pressure of 200 MPa or less, we are at the limit of UHPH and work more in the HPH range. In these conditions, the inactivation of PPOs is lower [20]. An intense inactivation of PPOs (>90%, [21]) has been observed at 300 MPa, and therefore, the oxidative damage to anthocyanins will be more intense, especially in a longer multipass process. The design shape and nanocoating of the valve and seat valve is another key aspect to enhance the mechanical effects and inactivation [20,31]. From our experience, working at 300 MPa in a single pass, the inactivation of PPOs, the antimicrobial effect, and the stability of anthocyanins are optimal, with strong protection of the color and better antioxidant activity [22]. In general, several works have observed a clear effect on particle size and viscosity after (U)HPH, which affects the extraction of molecules such as anthocyanins, phenolic aroma, and other nutrients, as well as the colloidal stability of the juice [62]. The effect on size is pressure-dependent, as observed by optical microscopy and light scattering [62] and AFM microscopy [22].

## 5. HHP vs. UHPH in the Extraction and Stability of Anthocyanins

The use of HHP has the advantage of allowing the processing of solids; therefore, it is possible to process grapes, grape pomace, or even bunches. The only limitation is that the size of the product to be treated must be smaller than the diameter and length of the vessel. Large HHP machines with a diameter of 0.38 m can be found on the market today. (https://www.hiperbaric.com/en/ (accessed on 24 August 2023)). Therefore, treating whole destemmed grape berries or bunches is perfectly possible on an industrial scale [25], as well as processing by-products such as pomace [26]. The extraction of solids can be obtained from the whole raw material and can be higher than in the case of UHPH, where a previous pre-extraction is needed. However, after the pre-extraction, the intense nano-fragmentation caused by UHPH in the colloids guarantees a better release of anthocyanins from the colloidal plant fragments [22]. In addition, the continuous nature of UHPH processing makes it more efficient from an industrial perspective.

Concerning anthocyanin oxidation, the main disadvantage of HHP is the variable effect on the inactivation of oxidative enzymes [16,27,32,33]. The protective effect of UHPH on anthocyanin color is due to its gentle action on them but also to the highly effective control of PPOs [21,23,56], which can be considered more intense than in HHP treatment, where inactivation is usually variable and under stronger conditions [64,65,66,67,68,69,70].

## 6. Environmental Impacts and Research Needs

Lastly, the UHPH technology is environmentally more sustainable than HHP due to the low consumption of water used for processing and cleaning. Sustainability is currently a key parameter in the wine industry [71,72,73,74,75]. The water consumption to produce 1 kg of grapes is about 550 L [76]. In a more recent study, the water food print for a 0.75 L bottle has 632 L [77]. Only in the industry, winemaking and cleaning consume in the range of 0.2–8 L of water per 1 L of wine, depending on several parameters, especially the size of the winery [73,74,78]. With UHPH, the need for cleaning products is lower because of the small volume of pipelines. Typical consumption is 750 L of water for 35 h of processing with 3000 L/h UHPH systems, which means less than 0.018 L of water per liter of product in a regular production cycle. Additionally, it has a lower consumption of energy and is not necessary for steam as processing or cleaning fluid. The main power consumption for UHPH is the motor of the UHPH pump, which is 11–18 kW for 60–150 L/h. The consumption for an industrial system of 3000 L/h is 290 kW/h. In terms of environmental impact, high-pressure extraction is superior to conventional methods and comparable to pulsed electric fields (PEFs), as reported in the literature [79], which is also another continuous emerging non-thermal technology used to increase the extraction of anthocyanins [80] and to accelerate the maceration of red wines [35,81,82,83,84,85,86]. Moreover, a better bioprotective capacity is obtained [87]. Therefore, UHPH and HHP are similar to PEF in anthocyanin protection and antioxidant activity [20,88]. All techniques accelerate the extraction of anthocyanins compared to conventional methods [81,82,83,84]. UHPH and PEF are continuous technologies and can, therefore, be applied to the product more efficiently [20,81]. HHPs and PEFs can be used in crushed grapes, but UHPH only in liquid juice [20,22].

At the same time, the commonly used packaging materials in HHP processing, i.e., polyethylene terephthalate, ethylene–vinyl alcohol copolymer, polyethylene, and polypropylene, could be a negative aspect of this extraction from a sustainability and environmental viewpoint, compared to other emerging extraction technologies such as UHPH [20], or moderate electric field [89].

Further enhancement of high-pressure processing sustainability could be a topic for future studies, which can be achieved by exploring the possibility of using biodegradable packaging materials or developing strategies for re-using and recycling high-pressure processing packaging material. Additionally, there are limited reports on combining high-pressure extraction and other techniques [90]. The possible effects of such combined approaches based on HPP and UHPH on anthocyanin extraction and stability could be considered in future research. At the same time, the bioavailability of the anthocyanins extracted by HPP and UHPH is among the information encouraged to be revealed by researchers [91]. This expected research can help provide information that can further promote the commercialization of HPP and UHPH so that the industry and consumers can benefit from high-pressure-based extracted anthocyanins.

## 7. Conclusions

High-pressure techniques are a powerful processing tool to enhance extraction in a gentle manner, protecting sensitive molecules such as anthocyanins and maintaining their stability during storage. The inactivation of oxidative enzymes and the absence of oxidation are key process parameters to achieve optimal conditions. 

As mentioned above, the main advantages of UHPH are its higher antimicrobial efficacy, better control of PPO enzymes, continuous processing, and low consumption of water, detergents, and energy. Conversely, the main disadvantage is the need to process liquid products and the impossibility of processing whole or crushed berries with skins. With regard to HHP, the most interesting processing feature is the possibility to process juice, crushed berries, or whole berries, which is optimal for the extraction of anthocyanins from skins to juice. The main disadvantage is a more variable behavior in the inactivation of enzymes, a less optimal batch process, and a higher consumption of water and energy.

These HP technologies open up new opportunities to obtain highly sensitive pigments with nutraceutical properties, such as anthocyanins, or to increase their content in juices, smoothies, purees, or derived beverages, even fermented as wines. The appropriate inactivation of oxidative enzymes, the high antimicrobial effect, and the gentle action allow us to reduce chemical additives such as SO_2_ and obtain food products with clean labels. 

## Figures and Tables

**Figure 1 antioxidants-12-01746-f001:**
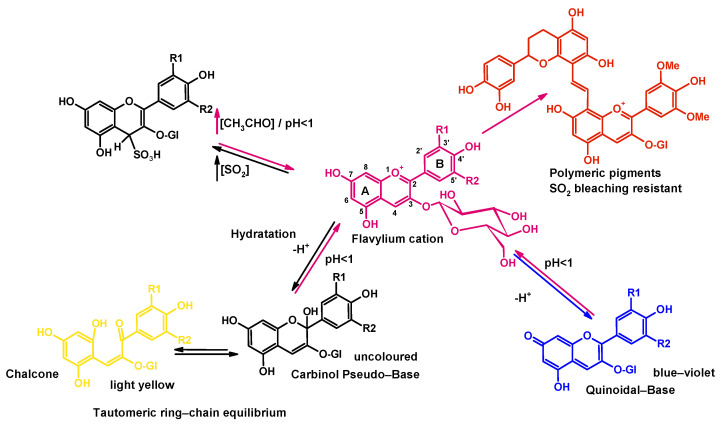
Reactions that can affect anthocyanins as a function of pH, hydration, SO_2_ concentration, and flavonoid content.

**Figure 2 antioxidants-12-01746-f002:**
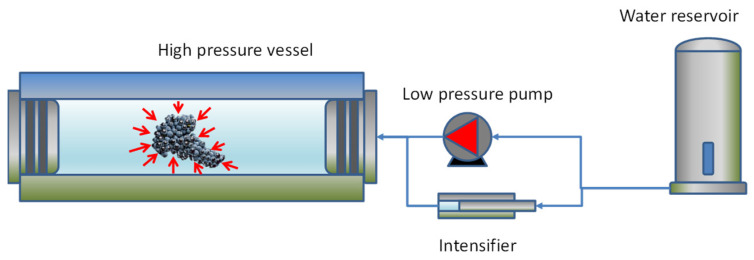
HHP system scheme with steel vessel, low-pressure pump, and high-pressure intensifier. Arrows around the grape cluster simulate the hydrostatic effect of the pressure.

**Figure 3 antioxidants-12-01746-f003:**
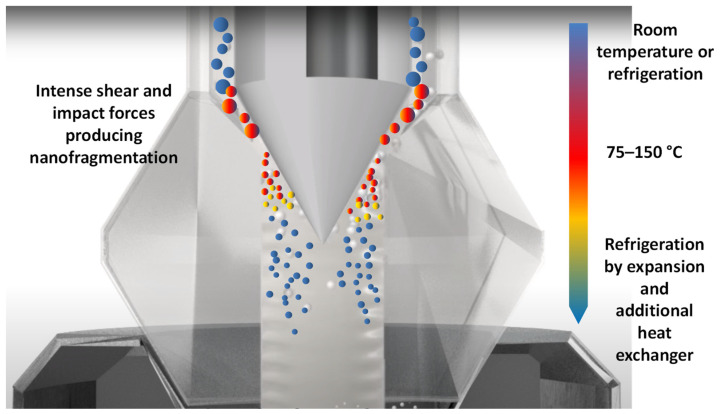
Effect of impact and shear forces on colloid nanofragmentation and temperature in a UHPH valve (adapted from http://www.ypsicon.com/ (accessed on 15 July 2023)).

**Figure 4 antioxidants-12-01746-f004:**
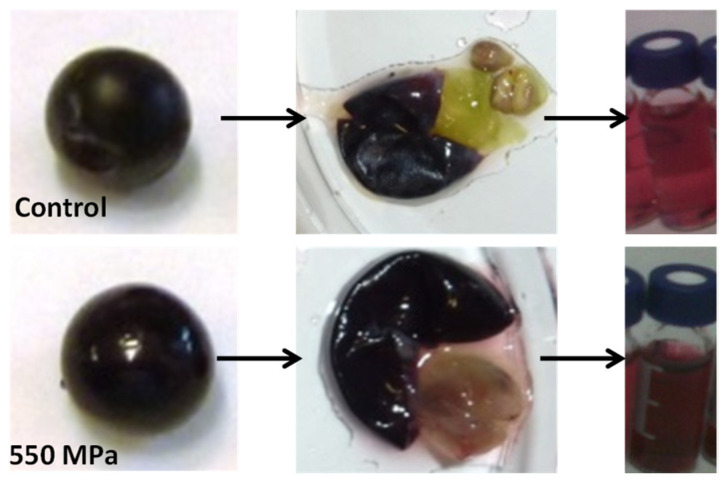
Effect of HHP on the surface and pulp of *Vitis vinifera* cv Tempranillo grape berries and on color extraction with a treatment of 550 MPa for 10 min.

**Figure 5 antioxidants-12-01746-f005:**
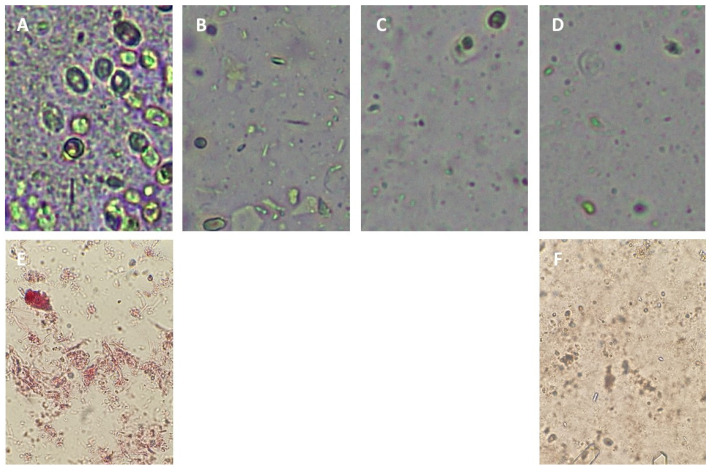
Optical microscopy of red juices from two varieties treated with UHPH, Tempranillo (*Vitis vinifera* L.) (**A**) Control 0.1 Mpa; (**B**) UHPH 240 Mpa; (**C**) UHPH 270 Mpa; (**D**) UHPH 300 MPa; Cabernet Sauvignon (*Vitis vinifera* L.); (**E**) Control 0.1 Mpa; (**F**) UHPH 300 MPa.

**Table 2 antioxidants-12-01746-t002:** Effect of different HHP treatments on the extraction and stability of anthocyanins in various fruits, juices, or by-products.

Fruit	Anthocyanins	HHP Parameters	Effects	References
Muscadine grape juice (*Vitis rotundifolia*)	Delphinidin, Petunidin, Peonidin and Malvidin 3,5-diglucosides (D35G, Pt35G, Pn35G, M35G)	400–550 ^1^ MPa, 15 min	Anthocyanin loss of 70% (400 MPa) and 46% at (550 MPa)	[42]
Merlot grape juice (*Vitis vinifera* L.)	C3G, Pt3G, Pn3G & M3G, acetyl C3G, acetyl Pn3G, acetyl M3G	600 MPa, 10 °C, 3 min	Higher anthocyanin content in HHP treatments compared to controls	[43]
Grape pomace from Dornfelder variety (*Vitis vinifera* L.)	D3G, C3G, Pt3G, Pn3G & M3G	200–600 MPa, ethanol (20–100%), 30–90 min, and temperature 20–70 °C	Higher extraction at 600 MPa, 50 °C, 100% ethanol (+23%) and related to the higher number of -OCH_3_ and -OH groups in the flavylium nucleus, extraction of M3G > Pn3G > Pt3G > D3G > C3G	[32]
Tempranillo grapes (*Vitis vinifera* L.)	D3G, C3G, Pt3G, Pn3G and M3G and acylated derivatives (acetyl, caffeoyl and *p*-coumaroyl)	200–550 MPa, 20 °C, 10 min	Migration of anthocyanins from skins to pulp and seeds. Increased extraction +80%. Higher concentration after fermentation.	[29]
Concord grape puree (*Vitis labrusca*)	Total monomers	600 MPa, 5 °C, 3 min,	Higher contents that the control and enough stability for 4 months	[39]
Raspberry (*Rubus idaeus*)	Cyanidin-3-glucoside (C3G) Cyanidin-3-sophoroside (C3S)	200–800 MPa, 18–22 °C, 15 min. Stored at: 4, 20, 30 °C for 9 days	Higher stability for 800 MPa stored at 4 °C	[44]
Strawberry (*Fragaria* × *ananassa*, cv. Elsanta)	Pelargonidin-3-glucoside (P3G) Pelargonidin-3-rutinoside (P3R)	200–800 MPa, 18–22 °C, 15 min. Stored at: 4, 20, 30 °C for 9 days	Higher stability for 800 MPa stored at 4 °C	[45]
Blackcurrant (*Ribes nigrum*)	Delphinidin-3-rutinoside (D3R) Cyanidin-3-rutinoside (C3R)	200–800 MPa, 18–22 °C, 15 min. Stored at: 5, 20, 30 °C for 7 days	Higher stability for 600–800 MPa stored at 5 °C	[46]
*Lonicera caerulea*	C3G, *p*-coumaroyl D3G, and D3R	200–600 MPa, 20 °C, 10 min	Anthocyanins protected, higher antioxidant activity	[33]
Blueberry pulp (*Vaccinium* spp.)	Delphinidin3-galactoside (D3Gal), D3G, Cyanidin-3-galactopyranoside (Cy3Gal), Delphinidin-3-arabinoside (D3A), C3G, Petunidin-3-galactoside (Pt3Gal); Peonidin-3-galactoside (Pn3Gal), Cyanidin-3-arabinoside (C3A), Pt3G, Pn3G, Malvidin-3-galactoside (M3Gal), Peonidin-3-arabinoside (Pn3A), Malvidin-3-arabinoside (M3A).	500 MPa, 5 min	HHP with low oxygen shows higher anthocyanin content and protective effect on color	[40]
Blueberry (*Vaccinium* spp.) puree	D3Gal, D3G, Cy3Gal, C3G, C3A, Petunidin-3-arabinoside (Pt3A), M3Gal, M3G, M3A	200–600 MPa, 20 °C, 20 min	Protective effect on color. At 300MPa was obtained the higher concentration of anthocyanins	[41]
Aronia (*Aronia melanocarpa*) berry purée	Total anthocyanins	200–600 MPa, 21–33 °C, 2.5–5 min	Preservation of anthocyanins, phenols and color. Antioxidant capacity unaffected	[38]
Strawberry (*Fragaria* × *ananassa* Duch.) cv. Senga Sengana purée and juice	Total monomeric anthocyanins	400–600 MPa, 20 °C, 1.5–3 min	Good stability of anthocyanins specially in juices	[47]

^1^ MPa: Mega Pascal.

**Table 3 antioxidants-12-01746-t003:** Effect of different (U)HPH treatments on the extraction and stability of anthocyanins in various fruits, juices, or by-products.

Fruit	Anthocyanins	UHPH Parameters	Effects	References
Cabernet sauvignon red must (*Vitis vinifera* L.)	D3G, C3G, Pt3G, Pn3G and M3G, acylated derivatives (acetyl, caffeoyl and *p*-coumaroyl), Vitisin B and Malvidin 3-glucoside vinylphenol (M3GvPh) and Malvidin 3-glucoside vinylguaiacol (M3GvG)	60 L/h, 300 ± 3 ^1^ MPa, inlet temperature 4 °C, in-valve 78 ± 2 °C (0.2 s) outlet temperature of 15 °C	Higher contents of anthocyanins in UHPH must with a selective protection of the acylated derivatives +9.3% with more red-bluish color Absence of anthocyanins in colloids.	[22]
Pomegranate (*Punica granatum* L.) juice	C3G, D3G, just optically evaluated OD_520nm_	100–150 MPa, inlet temperature 10 °C, 1–10 cycles, outlet max 42–46 °C	Not differences in color. In certain conditions higher polyphenol content and antioxidant activity than in the fresh juice.	[59]
Blackcurrant (*Ribes nigrum*) fruit juice	D3G, D3R, C3G and C3R	50–220 MPa, inlet temperature 4–20 °C, 1–5 passes	Slight reduction of anthocyanins. Preserve the bioactive and physicochemical quality.	[60]
Strawberry nectar	P3G	50–200 MPa, inlet temperature 25 °C, 1–5 passes, ΔT ≈ 19 °C/100 MPa	Anthocyanins and color slightly affected depending on number of cycles. Higher polyphenol content with multi passes at 200 MPa	[62]
Strawberry (*F. Ananassa*) juice	Color evaluation CieLab	60–100 MPa, inlet temperature 4–20 °C, 1–5 passes	Color, anthocyanins and antioxidant activity increases until 2 passes. Later degradation probably by thermal effect.	[63]

^1^ MPa: Mega Pascal.

## Data Availability

Not applicable.

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
