# Peer review of "Effect of HHP and UHPH High-Pressure Techniques on the Extraction and Stability of Grape and Other Fruit Anthocyanins"

_antioxidants, 2023, doi:10.3390/antiox12091746_

Round 1
Reviewer 1 Report
I suggest to expand article and add more examples (stability of anthocyjanins).
Author Response
I suggest to expand article and add more examples (stability of anthocyanins).
Thank you for your positive feedback and valuable comments. We have tried to expand several sections according to your suggestions. Please, note that we can find research on HHP and anthocyanins, but less work on UHPH and anthocyanins, more information on HPH, but of course working at lower pressure.
Reviewer 2 Report
The manuscript “Effect of HHP and UHPH high-pressure techniques on the extraction and stability of grape anthocyanins” by Morata et al., submitted to “Antioxidants”, focuses on a relevant topic in food science. The paper explores the use of High Hydrostatic Pressure (HHP) and Ultra-High Pressure Homogenization (UHPH) for extracting and stabilizing grape anthocyanins. The authors present a clear overview of these methods and discuss their benefits for anthocyanin extraction. The emphasis on non-thermal methods, which help preserve sensitive compounds, is a key point.
After reviewing the manuscript, I recognize the authors’ valuable work in this area of food science. However, some parts of the paper might benefit from some revisions. Making these changes could improve the paper’s clarity and strengthen its impact. I have listed my specific comments and suggestions below:
– The introduction does a good job of establishing the relevance and importance of anthocyanins. However, the paper might benefit from more concise language. For instance, instead of explaining every aspect of anthocyanins and high-pressure techniques in such detail, consider providing a general summary with the most critical points and then delving deeper into the subsequent sections. The introduction delves deep into specifics, such as the “Mach 3 for less than 0.2 seconds in valve”. While details are essential, the authors should determine if the introduction is the best place for such depth or if it would be better suited in a dedicated methods or technology section.
– The authors should ensure that terms remain consistent. For instance, if you start by calling it “High Hydrostatic Pressure (HHP)”, stick with this term or its abbreviation.
– Section 2 mentions “specificities regarding their characteristics and processing potential.” It might be useful to elaborate or briefly explain these specificities before diving into details.
– The section touches upon the potential applications of HHP and UHPH in wine processing. Consider expanding on this, possibly discussing any cost implications, scalability, or industry reception/adoption of these technologies. While the section does touch on some limitations, a more comprehensive discussion on the challenges or drawbacks of both HHP and UHPH might be beneficial for a balanced view.
– In section 3, the protective effect of anthocyanins with low oxygen pre-processing is an interesting point. It might be helpful to expand slightly on why this is the case or cite a reference that provides this explanation.
– The authors might ensure that all technical terms and acronyms are defined the first time they appear (i.e., acronyms like PPOs, C3G, D3G… might not be immediately clear to all readers).
– Section 4 begins abruptly. Starting with a brief statement introducing the significance or relevance of anthocyanins and their extraction would offer context to readers unfamiliar with the topic.
– While the authors have presented results from various studies, a brief discussion on their broader implications or contradictions between studies would add depth. For example, discuss why there might be a reduction in anthocyanins under certain conditions but not others.
– Whereas section 6 touches upon some environmental aspects like water consumption and energy use, elaborating more on the reasons behind these impacts can offer depth to the arguments.
– The mention of packaging materials in HHP is valuable. However, it might be beneficial to briefly elaborate on why these materials are less environmentally friendly than others or the alternatives used in UHPH.
– The comparison between high-pressure extraction, conventional methods, and pulsed electric fields could benefit from a bit more detail. For instance, in what aspects is one superior to the other?
– While conciseness is appreciated in a conclusion section, there’s room to elaborate more on specific advantages of HP technologies, especially if they were crucial points throughout the article. The conclusion should encapsulate the major points made in the paper. Ensure that the main points discussed in the prior sections are also mentioned here. For instance, if there was a comparison between HHP and UHPH in earlier sections, it might be beneficial to highlight the general superiority or difference in outcomes between these methods.
– Beyond just HHP vs. UHPH, how do these methods compare with other extraction or preservation methods in terms of efficiency, cost, scalability, and output quality? The authors may also include a discussion on the economic implications of adopting these technologies. This includes initial investment, operational costs, and potential ROI from improved product quality or yield.
– Every technology has its limitations. Discussing any drawbacks or challenges in implementing these techniques would provide a balanced view. While high-pressure techniques might inactivate certain harmful microorganisms, are there any safety concerns or unforeseen consequences of these processes?
– The manuscript’s citation style does not align with the journal’s preferred format. Ensuring that all references consistently adhere to the ‘Antioxidants’ citation style is crucial.
There are several instances throughout the manuscript where the language could be refined for clarity and fluency. I recommend a thorough proofreading by a native English speaker or professional editing service to ensure that the message is conveyed clearly.
Author Response
The manuscript “Effect of HHP and UHPH high-pressure techniques on the extraction and stability of grape anthocyanins” by Morata et al., submitted to “Antioxidants”, focuses on a relevant topic in food science. The paper explores the use of High Hydrostatic Pressure (HHP) and Ultra-High Pressure Homogenization (UHPH) for extracting and stabilizing grape anthocyanins. The authors present a clear overview of these methods and discuss their benefits for anthocyanin extraction. The emphasis on non-thermal methods, which help preserve sensitive compounds, is a key point.
Thank you.
After reviewing the manuscript, I recognize the authors’ valuable work in this area of food science. However, some parts of the paper might benefit from some revisions. Making these changes could improve the paper’s clarity and strengthen its impact. I have listed my specific comments and suggestions below:
Dear reviewer, thank you very much for your positive comments and valuable suggestions, we are grateful for the detailed review and the useful advice on how to improve the manuscript.
– The introduction does a good job of establishing the relevance and importance of anthocyanins. However, the paper might benefit from more concise language. For instance, instead of explaining every aspect of anthocyanins and high-pressure techniques in such detail, consider providing a general summary with the most critical points and then delving deeper into the subsequent sections. The introduction delves deep into specifics, such as the “Mach 3 for less than 0.2 seconds in valve”. While details are essential, the authors should determine if the introduction is the best place for such depth or if it would be better suited in a dedicated methods or technology section.
Thank you, according to your comment we have moved and adapted some specific paragraphs on UHPH from the Introduction to Section 2.
– The authors should ensure that terms remain consistent. For instance, if you start by calling it “High Hydrostatic Pressure (HHP)”, stick with this term or its abbreviation.
Sorry, but we think that we are consistent with HHP for high hydrostatic pressure and UHPH or (U)HPH for ultra high pressure homogenization. Sometimes we just use HPH for high pressure homogenization when the pressure is lower than 200 MPa. I have double checked it, but if you can find any mistake please tell us to correct it.
– Section 2 mentions “specificities regarding their characteristics and processing potential.” It might be useful to elaborate or briefly explain these specificities before diving into details.
Following your comments, we have added an introductory paragraph to identify and explain the specificities (L101-110).
– The section touches upon the potential applications of HHP and UHPH in wine processing. Consider expanding on this, possibly discussing any cost implications, scalability, or industry reception/adoption of these technologies. While the section does touch on some limitations, a more comprehensive discussion on the challenges or drawbacks of both HHP and UHPH might be beneficial for a balanced view.
According to your comments, we have included a new section on industrial scale-up and equipment (L128-142).
– In section 3, the protective effect of anthocyanins with low oxygen pre-processing is an interesting point. It might be helpful to expand slightly on why this is the case or cite a reference that provides this explanation.
Thank you, in this research work they prepare the berries by a pulping what is made with low or high air contact in the juice extract by processing with nitrogen atmosphere or normal air exposition. When they use nitrogen, the oxygen contents are very low and the degradation of anthocyanins after the HHP treatments is low, probably because of the inert conditions and the reduced effect of the residual PPOs activities on anthocyanins in such reductive conditions. We include this new discussion in the review.
– The authors might ensure that all technical terms and acronyms are defined the first time they appear (i.e., acronyms like PPOs, C3G, D3G… might not be immediately clear to all readers).
Thank you, even though we think that PPOs are quite common for readers interested in anthocyanins and enzymatic oxidation, the meaning is expanded in the abstract and table, and following your comment we have now also included it the first time it is used in the introduction. As for anthocyanins, there are a lot of glucosides and derivatives in Tables 2 and 3, and even though we think that these are well known to readers, we think that the abbreviations are expanded the first time they are used. Please, let us know if you think there is a need for further expansion.
– Section 4 begins abruptly. Starting with a brief statement introducing the significance or relevance of anthocyanins and their extraction would offer context to readers unfamiliar with the topic.
Thank you for this suggestion, we have added an introductory paragraph.
– While the authors have presented results from various studies, a brief discussion on their broader implications or contradictions between studies would add depth. For example, discuss why there might be a reduction in anthocyanins under certain conditions but not others.
Thank you. We explained that this is because the studies that reported a reduction used lower pressures than 300 MPa (in the range 50-200 MPa) and multi-pass mode, which produces more mechanical damage and probably more oxidation. We have now extended the discussion to show that at pressures below 200 MPa the inactivation of PPOs is much lower and therefore the oxidative damage is more intense. (L252-258). We have also included that this depends on the design of the valve.
– Whereas section 6 touches upon some environmental aspects like water consumption and energy use, elaborating more on the reasons behind these impacts can offer depth to the arguments.
Thank you, we have extended and improved the discussion according to your suggestions.
– The mention of packaging materials in HHP is valuable. However, it might be beneficial to briefly elaborate on why these materials are less environmentally friendly than others or the alternatives used in UHPH.
Thank you for your comment. It is not that they are less environmentally friendly than others, it is just that frequently in HHP the products are often processed in plastic pouches or bottles, and in UHPH a continuous process is not necessary.
– The comparison between high-pressure extraction, conventional methods, and pulsed electric fields could benefit from a bit more detail. For instance, in what aspects is one superior to the other?
Thank you for this suggestion. Although PEFs are not the main topic of this review, we have added some further comments on them. See L302-309.
– While conciseness is appreciated in a conclusion section, there’s room to elaborate more on specific advantages of HP technologies, especially if they were crucial points throughout the article. The conclusion should encapsulate the major points made in the paper. Ensure that the main points discussed in the prior sections are also mentioned here. For instance, if there was a comparison between HHP and UHPH in earlier sections, it might be beneficial to highlight the general superiority or difference in outcomes between these methods.
Thank you for your comment, we have added a new paragraph describing the main advantages and drawbacks of both technologies.
– Beyond just HHP vs. UHPH, how do these methods compare with other extraction or preservation methods in terms of efficiency, cost, scalability, and output quality? The authors may also include a discussion on the economic implications of adopting these technologies. This includes initial investment, operational costs, and potential ROI from improved product quality or yield.
Thank you for your comments, we wanted to focus on HHP vs UHPH. However, as suggested, we have included a new comparative discussion with PEFs as previously indicated. Moreover, we have added several paragraphs to the document concerning efficiency, cost, scale-up and quality. These have been highlighted in green along the manuscript.
– Every technology has its limitations. Discussing any drawbacks or challenges in implementing these techniques would provide a balanced view. While high-pressure techniques might inactivate certain harmful microorganisms, are there any safety concerns or unforeseen consequences of these processes?
Thank you for your suggestion. We have added new comments on the limitations and drawbacks of both technologies. In terms of unforeseen consequences, these technologies are very safe and we have described positive things regarding the absence or very low formation of thermal markers (HMF, furosine) and the absence of formation of toxic molecules such as ethyl carbamate.
– The manuscript’s citation style does not align with the journal’s preferred format. Ensuring that all references consistently adhere to the ‘Antioxidants’ citation style is crucial.
Thank you. Corrected.
There are several instances throughout the manuscript where the language could be refined for clarity and fluency. I recommend a thorough proofreading by a native English speaker or professional editing service to ensure that the message is conveyed clearly.
Thank you, we have checked it once again.
Reviewer 3 Report
The article should be of broad interest and should be a useful reference for those with a particular interest in the subject matter. The description of HHP and UHPH technologies is very clear. Although clearly written for the most part, a careful editing is required to correct occasional minor errors, e.g. missing words. Some needed minor corrections are listed in the comments which follow.
1. The title is somewhat limiting as information is presented on materials other than grapes.
2. There is some repetition and redundancy in the Abstract which should eliminated, thus providing room for other information.
3. Line 26 – “soft” is not a particularly descriptive term.
4. Lines 62-63 – “<3.4 water, <3.8 orange juice” should be written as “<3.4°C water, <3.8°C orange juice”.
5. Lines 106-107 – is it appropriate to describe >70°C as being a high temperature? (cf. Figure 3)
6. Line 141 – change “increases” to “affects”.
7. Lines 142-145 – this sentence is related to the authors’ own work – is there a reference for this? If not, treatment conditions should be stated.
8. Line 148 (also line 19) – is migration of anthocyanins to the seed an undesirable effect?
9. Line 196 – the raw juice should be identified.
10.Line 199 – should “(Figure 5c-d)” read “(Figure 5b-d)”?
11. Line 201 – AFM should be defined.
12. Line 220 – in the caption for Figure 5, E and F are identified as A and B.
Occasional minor edits are required. Any repetition should be eliminated.
Author Response
The article should be of broad interest and should be a useful reference for those with a particular interest in the subject matter. The description of HHP and UHPH technologies is very clear. Although clearly written for the most part, a careful editing is required to correct occasional minor errors, e.g. missing words. Some needed minor corrections are listed in the comments which follow.
Thank you for the positive comments and the valuable suggestions.
- The title is somewhat limiting as information is presented on materials other than grapes.
Thank you for this observation. Although we have chosen to focus on grapes, we have included interesting data from other fruits because the limited information available. Accordingly, we have slightly modified the title to include that as follows: Effect of HHP and UHPH high-pressure techniques on the extraction and stability of grape and other fruit anthocyanins.
- There is some repetition and redundancy in the Abstract which should eliminated, thus providing room for other information.
Thank you, we have revised it and removed a few words and fragments.
- Line 26 – “soft” is not a particularly descriptive term.
Thank you, now we have used “mild”.
- Lines 62-63 – “<3.4 water, <3.8 orange juice” should be written as “<3.4°C water, <3.8°C orange juice”.
Thank you, corrected according to your comment.
- Lines 106-107 – is it appropriate to describe >70°C as being a high temperature? (cf. Figure 3)
Thank you, even though the temperature inside the valve depends on the initial temperature of the valve and the amount and properties of the colloidal particles, we have written >75ºC in the text to be consistent with the figure.
- Line 141 – change “increases” to “affects”.
Thank you. The sentence has been corrected according to your comment.
- Lines 142-145 – this sentence is related to the authors’ own work – is there a reference for this? If not, treatment conditions should be stated.
Thank you, we have added a reference.
- Line 148 (also line 19) – is migration of anthocyanins to the seed an undesirable effect?
Not really, but it happens, and it has been explained in detail. The seeds have an adsorbent surface with an affinity for several compounds, including anthocyanins, which are retained on the surface. In fact, during the normal winemaking process, at the end, the seeds are coloured in the tank. With HHP, this process takes place even in the whole berry through the poration of the cells on the pulp and the migration of anthocyanins from the skins.
- Line 196 – the raw juice should be identified.
Thank you, it is described in the figure caption, and now in the text.
10.Line 199 – should “(Figure 5c-d)” read “(Figure 5b-d)”?
Thank you. Yes, you are right. The mistake has been corrected.
- Line 201 – AFM should be defined.
Thank you. Now it is defined in the text.
- Line 220 – in the caption for Figure 5, E and F are identified as A and B.
Thank you. Yes, you are right. The mistake has been corrected.
Occasional minor edits are required. Any repetition should be eliminated.
Thank you. We have revised it once again.
Round 2
Reviewer 2 Report
Thank you for addressing the comments thoroughly. The revisions enhance the clarity and depth of the manuscript. Given the improvements, I believe the manuscript is now better positioned for publication consideration.